# Epstein–Barr Virus-Positive Cutaneous and Systemic Plasmacytosis with TAFRO Syndrome-like Symptoms Successfully Treated with Rituximab

**DOI:** 10.3390/medicina59020216

**Published:** 2023-01-23

**Authors:** Seiji Kakiuchi, Hiroaki Akiyama, Isamu Harima, Ikumi Takagi, Junpei Rikitake, Yoko Kozuki, Mayumi Inaba, Hiroshi Fujiwara, Nozomu Kurose, Sohsuke Yamada, Yasufumi Masaki

**Affiliations:** 1Department of Hematology, Yodogawa Christian Hospital, 1-7-50, Kunijima, Higashi-Yodogawa-ku, Osaka 533-0024, Japan; 2Department of Pathology, Yodogawa Christian Hospital, 1-7-50, Kunijima, Higashi-Yodogawa-ku, Osaka 533-0024, Japan; 3Department of Respiratory Medicine, Yodogawa Christian Hospital, 1-7-50, Kunijima, Higashi-Yodogawa-ku, Osaka 533-0024, Japan; 4Department of Pathology and Laboratory Medicine, Kanazawa Medical University, 1-1, Daigaku, Uchinada, Kahoku 920-0293, Ishikawa, Japan; 5Department of Hematology and Immunology, Kanazawa Medical University, 1-1, Daigaku, Uchinada, Kahoku 920-0293, Ishikawa, Japan

**Keywords:** TAFRO syndrome, idiopathic multicentric Castleman’s disease, cutaneous and systemic plasmacytosis, Epstein–Barr virus, rituximab

## Abstract

Histopathologic findings in the lymph nodes of patients with thrombocytopenia, anasarca, fever, reticulin fibrosis, renal dysfunction, and organomegaly (TAFRO) syndrome are similar to those of idiopathic multicentric Castleman’s disease (iMCD), but TAFRO syndrome is different from iMCD in how it can progress rapidly and be fatal. These patients present scarce lymphadenopathy and low immunoglobulin levels. We present a case of cutaneous and systemic plasmacytosis (C/SP) that caused TAFRO syndrome-like symptoms which were successfully treated with rituximab. A 67-year-old woman presented with fever and a pruritic skin rash. Numerous plasma cells were observed in the peripheral blood and imaging revealed organomegaly, anasarca, and generalized lymphadenopathy. Subsequently, she rapidly developed thrombocytopenia as well as renal and heart failure. She tested positive for the Epstein–Barr virus (EBV), elevated immunoglobulins, and C/SP, which are also atypical for TAFRO syndrome, thereby complicating the diagnosis. However, after using the Japanese TAFRO Syndrome Research Group diagnostic criteria, we promptly administered rituximab to treat the C/SP with TAFRO-like symptoms and saved her life. Finally, histopathological observations of the lymph node biopsy helped confirm EBV-positive hypervascular-type iMCD. Therefore, diagnosing TAFRO-like syndromes based on the Japanese diagnostic criteria and following the associated treatment even without a confirmed diagnosis is crucial to improving the patient outcomes.

## 1. Introduction

Cutaneous and systemic plasmacytosis (C/SP) is a disease presenting multiple skin eruptions, lymphadenopathy, and polyclonal hypergammaglobulinemia. Owing to its skin involvement, C/SP is often compared to idiopathic multicentric Castleman disease (iMCD) [1,2]. TAFRO syndrome was first described by Takai et al. in 2010 and is characterized by thrombocytopenia, anasarca, fever, renal failure or reticulin fibrosis, and organomegaly [3]. Histopathologically, the lymph nodes in TAFRO syndrome are classified as mixed, plasmacytoid, or hypervascular according to those in iMCD [4]. Based on the histological similarity of the lymph nodes, Fajgenbaum et al. categorized TAFRO syndrome as a subtype of iMCD [5]. Additionally, Iwaki et al. proposed diagnostic criteria for iMCD-TAFRO, which requires histopathological manifestations of the lymph nodes in addition to clinical symptoms [6]. However, the clinical features of the most common form of iMCD, iMCD-not otherwise specified (NOS), differ significantly from those of TAFRO syndrome [7]. iMCD-NOS usually develops slowly and presents with thrombocytopenia and hypergammaglobulinemia. In contrast, TAFRO syndrome often exacerbates rapidly and presents as thrombocytopenia without hypergammaglobulinemia [6,7]. Furthermore, performing a biopsy procedure in a patient with TAFRO syndrome is often challenging due to the small size of the lymph nodes, poor general condition, or severe thrombocytopenia [8]. As most cases are severe and require immediate treatment to ascertain survival rates, the Japanese TAFRO Syndrome Research Group proposed diagnostic criteria that do not require a histological analysis of the lymph nodes [9,10]. However, only limited cases are resolved with the recommended primary steroid therapy alone and second-line therapies such as tocilizumab, rituximab, and cyclosporine are often required. However, the precise and optimal treatment strategy remains unclear [11].

In this report, we describe a case of C/SP with TAFRO-like syndrome with an Epstein–Barr virus (EBV) infection that met the Japanese diagnostic criteria for TAFRO syndrome. We excluded iMCD-TAFRO because of hypergammaglobulinemia. Despite the rapidly progressing deterioration with C/SP and myocarditis, we successfully treated the patient with rituximab.

## 2. Case Report

A 67-year-old Japanese woman was admitted to our hospital after one week of a remittent fever exceeding 37 °C and systemic erythema with pruritus, which did not improve after treatment with acetaminophen and topical steroids. On initial examination, her temperature was 38.5 °C, her blood pressure was 90/60 mmHg, her pulse rate was 110 beats/min, and her SpO_2_ was 96% without oxygen support. The blood tests indicated a high inflammatory response with a white blood cell count of 11,700 cells/μL and a C-reactive protein level of 8.24 mg/dL, renal dysfunction with urea nitrogen at 27.3 mg/dL, creatinine at 1.31 mg/dL, and an estimated glomerular filtration rate of 31.9 mL/min/1.73 m^2^. The plain computed tomography (CT) imaging revealed numerous lymphadenopathies, with a maximum diameter of approximately 8 mm, generalized from the neck to the inguinal region. Although the repeated blood cultures were negative, the Sequential Organ Failure Assessment score was six, so we initially administered 1 g of intravenous (IV) meropenem every 12 h and noradrenaline to stabilize her blood pressure because we suspected septic shock. However, the plasma cell number was visibly increased in the peripheral blood and the generalized erythema worsened. Thus, meropenem was discontinued to avoid the possibility of drug eruption. After six days of admission, the platelet and fibrinogen levels decreased, leading to disseminated intravascular coagulation and requiring the administration of fresh-frozen plasma (FFP). The blood tests revealed positive EBV DNA and elevated soluble interleukin-2 receptor and interleukin-6 levels. The polyclonal immunoglobulins were also elevated (except M-protein) and a free light chain restriction was not observed (Table 1). Contrast-enhanced chest and abdominal CT revealed the appearance of bilateral pleural effusions and ascites. Compared with that during the initial examination, the size of the lymphadenopathy remained unaltered, whereas the liver, spleen, and kidneys appeared to be swollen (Figure 1A–D). A bone marrow biopsy was performed on the same day, which appeared to be normocellular with no increase in the megakaryocytes or reticulin fibers and MF-0 fibrosis. Plasma cells accounted for approximately 30% of the nucleated cells, and eosinophils and histiocytes were also abundant. Megakaryocytes with nuclear atypia exhibiting hypersegmentation were also observed (Figure 2A,B). The flow cytometry did not reveal any light chain-restricted or abnormal cells, and the G-band examination indicated a normal karyotype. The generalized erythema with pruritus did not improve despite the topical use of steroids (Figure 3A). The skin biopsy revealed dilated small blood vessels in the papillary dermis with a surrounding infiltration of the lymphocytes, plasma cells, and histiocytes with edema (Figure 3B,C). Although no tick bites were observed, we considered a possible rickettsial infection and started a levofloxacin and minocycline treatment which was administered for eight days. However, this treatment was terminated when the PCR results were negative for rickettsiosis. The general condition and blood coagulation abnormalities of the patient continued to worsen; therefore, she was continued on platelet and FFP transfusions. Additionally, she was treated with high-dose steroid therapy (methylprednisolone 500 mg/day for three days) to treat the generalized indurated edema. This treatment resulted in a decrease in the percentage of plasma cells in the peripheral blood. Echocardiography indicated diffusely left ventricular hypokinesis with a 20–30% ejection fraction and pericardial effusion. Consequently, the patient was started on dobutamine (4.5 μg/kg·min). Furthermore, the renal function and thrombocytopenia deteriorated gradually, fulfilling the Japanese diagnostic criteria for TAFRO syndrome at this point. Therefore, we started a rituximab treatment at a dose of 375 mg/m^2^ and subsequently repeated it every seven days. The fever and the skin rash rapidly disappeared; the renal failure, heart failure, and blood coagulation abnormalities improved; and the patient no longer depended on transfusions. Upon stabilization, we performed inguinal lymph node and myocardial biopsies. The inguinal lymph node biopsy was negative for human herpesvirus-8 latency-associated nuclear antigen (Figure 4A). The histopathological analysis revealed a hypervascular type of iMCD. The lymphoid follicles were markedly atrophic, with dense hyper endothelial vessels between the follicles (Figure 4B). CD23 staining barely identified lymphoid follicles. The network of follicular dendritic cells was disorganized, with follicular dendritic cells extending outside the germinal center (Figure 4C). The perivascular area was infiltrated with CD38-positive plasma cells and small lymphocytes (Figure 4D). The immunoglobulin G4/CD38 ratio in the plasma cells was less than 1%. The immunoglobulin heavy chain and T-cell receptor rearrangements were negative. Furthermore, the EBV-encoded RNA in situ hybridization (EBER-ISH) was positive for an infection (Figure 4E). The myocardial biopsy exhibited atrophic myocardial fibers, edema between myocardial fibers, myocardial fiber degeneration, and hemorrhage. No fibrosis was evident in the interstitium but it was accompanied by a mild lymphocytic and histiocytic infiltrate (Figure 5A). The EBER-ISH was negative in this sample (Figure 5B). Since the patient developed a fever even after the rituximab treatment, we started the patient on 2 g of IV cefepime every 12 h and switched to 1 g of IV meropenem every 12 h, with no resolution of the fever. The plasma cytomegalovirus DNA was later found to be positive, thus 5 mg/kg of IV ganciclovir every 12 h was concurrently added, which successfully relieved the fever. The patient was discharged 50 days after their admission. Finally, she completed eight cycles of rituximab treatment in the outpatient ward and remained alive with a good performance as of December 2022. Figure 6 depicts the complete clinical course of the patient during hospitalization.

## 3. Discussion

We have reported a case of TAFRO-like syndrome with hypergammaglobulinemia and C/SP. The patient presented iMCD-like symptoms, such as thrombocytopenia, anasarca, fever, renal dysfunction, organomegaly, and histopathology of the fine lymph nodes, with a highly elevated inflammatory response consistent with the diagnostic criteria for TAFRO syndrome. However, the presence of elevated polyclonal immunoglobulin and the absence of elevated alkaline phosphatase levels were atypical. Additionally, an EBV infection was detected in the serum and histopathology of the lymph nodes. An association between EBV and iMCD has been previously suggested, with 17 of 19 EBV-positive iMCD cases reported to be histological of the hypervascular type [12]. In line with these results, the lymph node histopathology results were consistent with that of hypervascular iMCD, indicating a possible relationship between EBV and iMCD. Consequently, this complex case could not be easily categorized as a TAFRO syndrome owing to its atypical symptoms.

Additionally, C/SP has been previously reported as a skin lesion related to iMCD [1,2]. The skin biopsy of the patient indicated C/SP, which is considered to be a related histological change. The examination of the bone marrow revealed plasmacytosis, suggesting a systemic inflammatory response and megakaryocytes with atypia, indicating a potential extranodal lesion with findings resembling TAFRO syndrome [13]. The myocardial biopsy was negative for EBV via EBER-ISH, indicative of nonspecific lymphocytic myocarditis. TAFRO syndrome with cardiomyopathy has been reported in a few cases, and this case may have been a related manifestation [14,15,16].

In this case, it is important to distinguish thrombotic thrombocytopenic purpura (TTP), hemolytic uremic syndrome (HUS), and Kaposi Sarcoma Inflammatory Cytokine Syndrome (KICS). While TTP and HUS are similar to this case in that they cause fever, thrombocytopenia, acute kidney injury, and TTP, these were excluded because the ADAMTS13 activity was not decreased. HUS was also ruled out because of the absence of hemolytic anemia and gastrointestinal symptoms such as diarrhea or bloody stools. KICS is a recently proposed disease entity that occurs in patients simultaneously infected with human immunodeficiency virus (HIV) and human herpesvirus-8 (HHV-8) [17,18]. KICS is clinically indistinguishable from iMCD-TAFRO due to the presence of varying degrees of lymph node swelling, pancytopenia, and systemic inflammatory syndromes. Therefore, differential diagnosis requires a bone marrow examination and lymph node biopsy. In the present case, KICS was ruled out because the lymph node biopsy was consistent with iMCD-TAFRO and the patient had no evidence of a HIV or HHV-8 infection.

In conclusion, the histopathology of the lymph nodes resembled iMCD, and the biopsy results of the other organs were also considered to be relevant findings. This case was atypical for TAFRO syndrome and presented a diagnostic challenge. TAFRO-like syndrome was diagnosed based on the Japanese criteria without the requirement for a lymph node biopsy. After the failure of steroid therapy, an early second-line treatment with rituximab was successful. Rituximab was reported to be a promising option based on a multicenter retrospective study that compared the efficacy of rituximab and cyclosporine A as a second-line therapy for TAFRO syndrome in Japan [11]. Van Rhee et al. also reported that a rituximab treatment was relatively effective for iMCD-TAFRO [19]. Moreover, individual cases with iMCD or TAFRO syndrome with cardiomyopathy have been reported with the following treatments to improve cardiomyopathy: tocilizumab, tacrolimus, rituximab, steroids, CHOP-like chemotherapy, and rituximab plus CHOP-like chemotherapy [13,14,15,20,21]. Therefore, rituximab is an effective treatment option for TAFRO-like syndromes such as the present case.

Even in atypical cases of TAFRO-like syndrome, as in the case of our patient, a prompt diagnosis based on the Japanese diagnostic criteria is crucial to proceed with the recommended treatment for TAFRO syndrome.

## Figures and Tables

**Figure 1 medicina-59-00216-f001:**
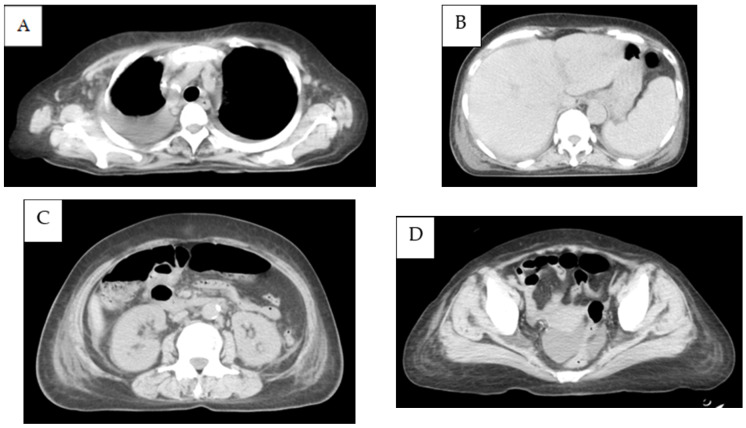
Contrast-enhanced chest and abdominal computed tomography recorded on day 6 of admission. (**A**) Subtle axillary lymph node enlargement and pleural effusion; (**B**) exacerbated hepatosplenomegaly; (**C**) enlarged kidney and small para-aortic lymphadenopathy; (**D**) ascites effusion.

**Figure 2 medicina-59-00216-f002:**
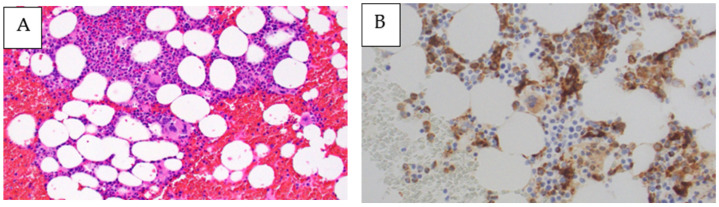
Bone marrow biopsy collected on day 7 of admission. (**A**) Hematoxylin and eosin staining indicating a normocellular bone marrow. No increase in megakaryocytes or reticulin fibers. Plasma cells and eosinophils are hyperplastic. Megakaryocytes with hypersegmented nuclear atypia are present Magnification ×400; (**B**) CD68 staining depicts an increased number of histiocytes. Magnification ×400.

**Figure 3 medicina-59-00216-f003:**
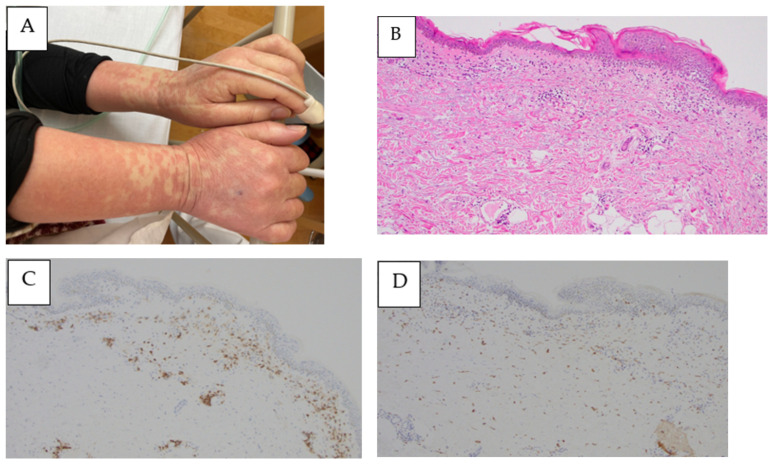
(**A**) Pruritic erythema on both forearms. Skin biopsy on day 7 of admission indicates: (**B**) small vessels dilated in the papillary dermis with edema by hematoxylin and eosin staining. Magnification ×100; (**C**) perivascular infiltration of plasma cells by CD38 staining. Magnification ×100; (**D**) perivascular infiltration of histiocytes by CD68 staining. Magnification ×100.

**Figure 4 medicina-59-00216-f004:**
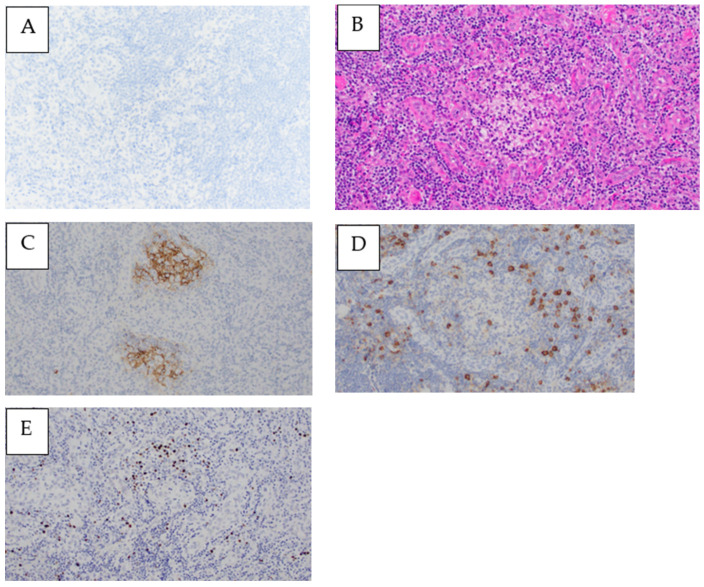
Inguinal lymph node biopsied on day 18 of admission. (**A**) Human herpesvirus-8 latency-associated nuclear antigen-1 (HHV-8 LANA-1) was negative in lymph node specimens. Magnification ×200; (**B**) hematoxylin and eosin staining demonstrating lymphoid follicles markedly atrophic, with a dense vascular proliferation of high endothelial venules between the follicles. Magnification ×200; (**C**) CD23 staining displaying barely visible lymphoid follicles, with a disorganized network of follicular dendritic cells extending beyond the pre-existing germinal center. Magnification ×200; (**D**) CD38 staining showing infiltration of positive plasma cells and small lymphocytes around blood vessels. Magnification ×200; (**E**) scattered cells were positive for Epstein–Barr virus-encoded RNA in situ hybridization (EBER-ISH). Magnification ×200.

**Figure 5 medicina-59-00216-f005:**
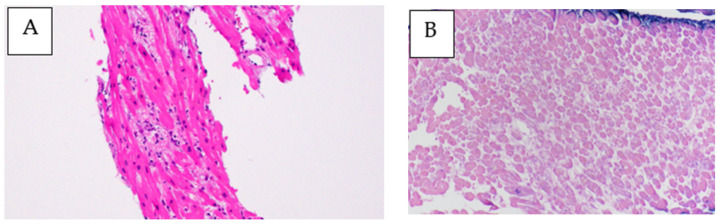
Myocardial biopsy on day 19 of admission. (**A**) Atrophic myocytes with myocardial degeneration and hemorrhage by Hematoxylin and Eosin staining. Fibrosis is not apparent in the interstitium, along with a mild infiltrate of lymphocytes and histiocytes. Magnification ×200; (**B**) Negative Epstein–Barr virus-encoded RNA in situ hybridization (EBER-ISH) analysis. Magnification ×200.

**Figure 6 medicina-59-00216-f006:**
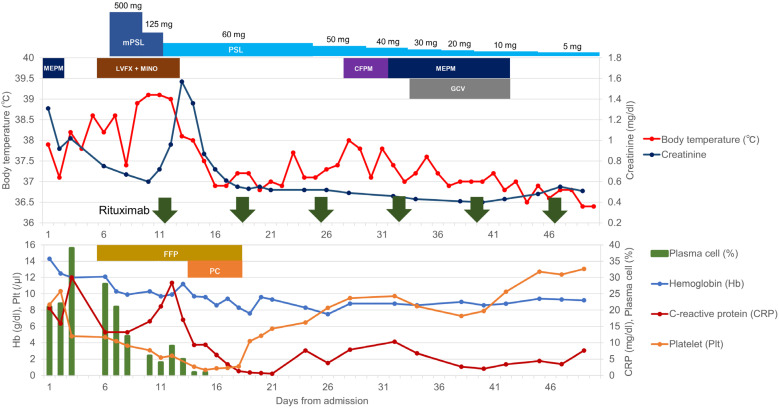
Clinical course from patient admission to discharge. Abbreviations: CFPM, cefepime; DOB, dobutamine; FFP, fresh-frozen plasma; GCV, ganciclovir; LVFX, levofloxacin; MINO, minocycline; mPSL, methyl prednisolone; MEPM, meropenem; NAD, noradrenaline; PC, platelet concentrate; PSL, prednisolone.

**Table 1 medicina-59-00216-t001:** Blood laboratory findings on day 6 of admission.

<Complete Blood Count>			<Others>		
White blood cell	13,300	μ/L	IgG	2558	mg/dL
Neutrophil	34	%	IgA	334	mg/dL
Lymphocyte	13	%	IgM	1630	mg/dL
Monocyte	2	%	IgD	0.7	mg/dL
Eosinophil	22	%	IgE	1490	IU/mL
Myelocyte	1	%	IgG4	54.8	mg/dL
Plasma cell	28	%	κ free light chain	207	mg/dL
Hemoglobin	12.1	g/dL	λ free light chain	211	mg/dL
Red blood cell	473	×10^4^/μ/L	Soluble interleukin-2 receptor	6120	U/mL
Hematocrit	40.2	%	Interleukin-6	13.2	pg/mL
Mean corpuscular volume	85	fl	VEGF (plasma)	28	pg/mL
Platelet	11.7	×10^9^/L	Proteinase 3-ANCA	<1.0	U/mL
			Myeloperoxidase-ANCA	<1.0	U/mL
<Coagulation>			Antinuclear antibody	<40	
PT	50	%	Anti-ds DNA antibodies IgG	<10	
APTT	37.6	sec	Anti-SS A	1.6	U/mL
PT–INR	1.63		Anti-SS B	5.2	U/mL
Fibrinogen	101	mg/dL	Rheumatoid factor	5.1	IU/mL
D-dimer	1.73	μg/mL	HIV antigen/antibody	(−)	
FDP	6.8	μg/mL	Hepatitis B surface antigen	(−)	
Antithrombin III	64	%	Hepatitis B surface antibody	(+)	
			Hepatitis B virus DNA	(−)	
<Blood Chemistry>			Hepatitis C virus antibody	(−)	
Total protein	6.5	g/dL	T-SPOT. TB	(−)	
Albumin	1.7	g/dL	EB-VCA-IgG	20	
Aspartate aminotransferase	18	U/L	EB-VCA-IgM	<10	
Alanine aminotransferase	19	U/L	EB-EADR IgG	<10	
Alkaline phosphatase	90	U/L	EB-EADR IgM	<10	
γ-Glutamyltranspeptidase	40	U/L	EBNA	10	
Lactate dehydrogenase	388	U/L	Epstein–Barr virus DNA	4.69	Log IU/mL
Total bilirubin	0.77	mg/dL	Cytomegalovirus DNA	(−)	
Blood urea nitrogen	19.1	mg/dL	Human herpesvirus-8 DNA	(−)	
Uric acid	8.0	mg/dL	ACE	10.0	U/L
Creatinine	0.75	mg/dL	ADAMTS13 activity	60	%
Sodium	138	mEq/L			
Potassium	3.5	mEq/L	<Dipstick urinalysis>		
Chloride	105	mEq/L	pH	5.0	
Phosphate	3.7	mg/dL	Specific gravity	1.023	
Magnesium	2.0	mg/dL	Blood	(++)	
Creatinine phosphokinase	26	U/L	Protein	(++)	
C-reactive protein	5.29	mg/dL	<Urine microscopy>		
Haptoglobin	143	mg/dL	White blood cell	5–9	/HPF
			Red blood cell	30–49	/HPF

Abbreviations: ACE, angiotensin-converting enzyme; ANCA, antineutrophil cytoplasmic antibody; APTT, activated partial thromboplastin time; DNA, deoxyribonucleic acid; ds, double strand; EA, early antigen; DR, diffuse and restrict complex; EB, Epstein–Barr virus; EBNA, Epstein–Barr virus nuclear antigen; FDP, fibrin/fibrinogen degradation products; HIV, human immunodeficiency virus; HPF, high power field; Ig, immunoglobulin; PT, prothrombin time; PT-INR, PT-international normalized ratio; SS, Sjögren’s-syndrome-related antigen; VCA, virus capsid antigen; VEGF, vascular endothelial growth factor.

## Data Availability

All experimental data to support the findings of this study are available by contacting the corresponding author.

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
