# Peer review of "Epstein–Barr Virus-Positive Cutaneous and Systemic Plasmacytosis with TAFRO Syndrome-like Symptoms Successfully Treated with Rituximab"

_medicina, 2023, doi:10.3390/medicina59020216_

Round 1

Reviewer 1 Report

This is a complex scenario where the authors have successfully diagnosed and treated patient who developed constellation of the symptoms resembling TAFRO/ MCD syndrome.  The patient presented with plethora of the signs and symptoms and posed a very hard differential diagnostic challenge. 

The authors are to be congratulated on excellent care that they provided to the patient and for reaching the positive outcome with Rituximab ifusions.

I have enjoyed reading this interesting case very much.  I do have the following comments that I think would improve the educational value of the paper.

1.       In discussion section please mentioned other entities that are relatively recently described such as Kaposi sarcoma inflammatory cytokines syndrome ( https://pubmed.ncbi.nlm.nih.gov/33268763/). This syndrome has almost identical clinical features and this must be highlighted;

2.       Additionally- thrombocytopenia, fever, low fibrinogen level and elevated D-dimer are consistent with disseminated intravascular coagulopathy.  This is another differential diagnosis that authors failed to discuss.

3.       Finally, the symptoms of fever, acute kidney injury, thrombocytopenia might be seen in patients with microangiopathic hemolytic anemia such as TTP and HUS and this needs to be mentioned in the discussion as well.

Author Response

Response to Reviewer

To Referee: 1

Comment 1.

In discussion section please mentioned other entities that are relatively recently described such as Kaposi sarcoma inflammatory cytokines syndrome ( https://pubmed.ncbi.nlm.nih.gov/33268763/). This syndrome has almost identical clinical features and this must be highlighted;

Comment 3.

Finally, the symptoms of fever, acute kidney injury, thrombocytopenia might be seen in patients with microangiopathic hemolytic anemia such as TTP and HUS and this needs to be mentioned in the discussion as well.

Response 1&3.

Thank you for the reference suggestion and for mentioning the additional symptoms and diseases. As you pointed out, Kaposi sarcoma inflammatory cytokines syndrome is consistent with the clinical findings in this case, and I added to the Discussion that lymph node biopsy is mandatory for differentiation. In addition, we have also included discussion on TTP and HUS because of the need to differentiate them based on the concordance of their clinical findings. We have described the need for a differential diagnosis in paragraphs 164 to 176 on page 4.

Comment 2.

 Additionally- thrombocytopenia, fever, low fibrinogen level and elevated D-dimer are consistent with disseminated intravascular coagulopathy.  This is another differential diagnosis that authors failed to discuss.

Response 2.

Thank you for your insightful comments. We agree with the need to state that this case was a disseminated intravascular coagulation condition based on blood test findings. We included this statement on page 2, lines 87 to 90.

Author Response

Response to Reviewer

To Referee: 2

Comments concerning Title part.

Title: Please rephrase it, since it is unclear.

Response.

Thank you for this suggestion.

I rephrased the title to ‘’Epstein-Barr virus-positive cutaneous and systemic plasmacytosis with TAFRO syndrome-like symptoms successfully treated with rituximab'’.

Comments concerning Abstract part.

Line 24: I think “differ” is not the correct verb. Also, the entire sentence need to be rephrased.

Line 26: Remove “by”

Line 27: Remove “a”

Response.

Line 24: I think “differ” is not the correct verb. Also, the entire sentence need to be rephrased.

→Thank you for pointing this out. I have edited the entire sentence(page 1, lines 23 and 24). The sentence now reads: Histopathologic findings in the lymph nodes of patients with TAFRO syndrome are similar to those of idiopathic multicentric Castleman's disease (iMCD), but TAFRO syndrome is different from iMCD in that it can progress rapidly and be fatal.

Line 26: Remove “by”

Line 27: Remove “a”

→Thank you for pointing these errors out. I have removed these words.

Comments concerning Case report part.

Case report

Line 72: Please change “flaccid” with another term

Line 78: Please add eGFR

Line 81: Why did you suspect septic shock? Have you performed a SOFA score? Had you collected blood cultures? Explain please and add more details because in this form your suspicion isn’t justified. Add meropenem dose.

Line 86: Please clarify what do you mean for “the presence of EBV”

Line 98: “the generalized erythema with pruritus did not improve” please explain why it should have improved? What treatment did you administer?

Line 102: Why did you administer levofloxacin plus minocycline? It was not sufficient only one drug?

Lines 130-132: Please explain why you used antibiotics and antivirals and what drugs. It is unclear for me.

Response.

Line 72: Please change “flaccid” with another term

→ I have replaced "flaccid" with ''remittent'' as shown on page 2, lines 72 and 73.

Line 78: Please add eGFR

→ Great idea. I have stated on page 2, line 79, that the eGFR is 31.9 ml/min/1.73 m2.

Line 81: Why did you suspect septic shock? Have you performed a SOFA score? Had you collected blood cultures? Explain please and add more details because in this form your suspicion isn’t justified. Add meropenem dose.

→ Thank you for these great questions. We suspected septic shock because of the SOFA score of 6. The repeat blood cultures were negative. We included this information on page 2, lines 81-84. We also included the dose of meropenem.

Line 86: Please clarify what do you mean for “the presence of EBV”

→ We mean that the blood test was positive for EBV DNA. We added this clarification to page 2, lines 89 and 90,

Line 98: “the generalized erythema with pruritus did not improve” please explain why it should have improved? What treatment did you administer?

→ Thank you for these questions. The generalized erythema did not improve despite the use of topical steroids. We included this statement on page 3, lines 101 and 102.

Line 102: Why did you administer levofloxacin plus minocycline? It was not sufficient only one drug?

→ It is possible that only levofloxacin or minocycline would have been sufficient as you indicated. However, we administered both to decrease the possibility of severe rickettsial infection.

Lines 130-132: Please explain why you used antibiotics and antivirals and what drugs. It is unclear for me.

→ Thank you for this suggestion. We have provided details on the administration of antimicrobials and antivirals when a patient develops a fever after rituximab treatment, on page 3, lines 133-137.

Comments concerning Discussion part.

I suggest adding a table or an image to resume the difference between the syndromes you described, in order to help readers to better understand what you are saying.

Moreover, I suggest better formatting the table 1, since it is difficult to read.

Figure 6 could be better presented.

Response.

Thank you for this suggestion. We have revised Table 1 and Figure 6 to make them more understandable.

Round 2

Reviewer 1 Report

I am pleased with revised version and I believe the paper has been sufficiently improved to be accept in the present form. Congratulations! 

Author Response

Thank you for reviewing our case report. We could improve the article thanks to your advice.

Reviewer 2 Report

Thank you to fix and change what I suggested.

English language improved, however I suggest revising one more time to make it more fluid and easily understandable.

I have only minor concerns:

Line 24: "in that", better to say "in which"

Line 132: "Since the patient developed fever" (without "a")

As regards Kaposi sarcoma, I suggest this interesting reference: 10.3390/idr14020028

Kind regards

Author Response

Response to Reviewer

To Referee: 2

Comments.

Line 24: "in that", better to say "in which"

Line 132: "Since the patient developed fever" (without "a")

As regards Kaposi sarcoma, I suggest this interesting reference: 10.3390/idr14020028

Response.

Line 24: "in that", better to say "in which"

Line 132: "Since the patient developed fever" (without "a")

→Thank you for pointing these errors out. I have corrected the words according to your advice.

As regards Kaposi sarcoma, I suggest this interesting reference: 10.3390/idr14020028

→ Thank you for suggesting an interesting reference. We have added your reference as a citation for KICS on page 5, line 169.

Round 3

Reviewer 2 Report

Thank you for resubmitting the improved version.